# Effectiveness of Integrated Technology Apps for Supporting Healthy Food Purchasing and Consumption: A Systematic Review

**DOI:** 10.3390/foods10081861

**Published:** 2021-08-12

**Authors:** Sook Yee Lim, Kai Wei Lee, Wen-Li Seow, Nurul Azmawati Mohamed, Navin Kumar Devaraj, Syafinaz Amin-Nordin

**Affiliations:** 1Department of Medical Microbiology, Faculty of Medicine and Health Sciences, Universiti Putra Malaysia, Serdang 43400, Selangor, Malaysia; l.sookyee@yahoo.com (S.Y.L.); wenlyseow@gmail.com (W.-L.S.); 2Faculty of Applied Sciences, UCSI University, Cheras 56000, Wilayah Persekutuan Kuala Lumpur, Malaysia; 3Department of Pre-Clinical Sciences, Faculty of Medicine and Health Sciences, Universiti Tunku Abdul Rahman, Kajang 43000, Selangor, Malaysia; lee_kai_wei@yahoo.com; 4Department of Basic Medical Sciences 2, Faculty of Medicine and Health Sciences, Universiti Sains Islam Malaysia, Persiaran Ilmu, Bandar Baru Nilai, Nilai 71800, Negeri Sembilan, Malaysia; drnurul@usim.edu.my; 5Department of Family Medicine, Faculty of Medicine and Health Sciences, Universiti Putra Malaysia, Serdang 43400, Selangor, Malaysia; knavin@upm.edu.my

**Keywords:** healthy food intake, food purchase, smartphone applications, efficacy, healthy eating, literature searches

## Abstract

A healthy diet is essential for good health and nutrition, though literature showed that there are various factors affecting the intention to purchase and consume healthy food. Technology integration is known to be useful in various aspects, but findings from studies on the efficacy of technology integration to improve healthy food consumption and purchase have largely been inconsistent. Therefore, we aimed to examine the efficacy of interventions that use technology apps to improve healthy food purchasing and consumption in adults. Relevant studies were identified through PubMed, Scopus, CINAHL, SportDiscuss and ACM Digital Library. Twenty studies were included in the systematic review. The majority of studies (*n* = 18) used a smartphone in the intervention, and only two studies used a personal digital assistant. The results showed that technology integration-based intervention favoured healthy changes in household food purchases, and increased consumption of healthy food and healthy eating outcomes – albeit to different extents. Overall, technology apps are convenient and user-friendly tools to encourage a change in healthy food purchase and consumption among people.

## 1. Introduction

Healthy food purchases and consumption are the trending topics among consumers nowadays. An increase in healthy food purchasing patterns may motivate consumers to practice healthier food consumption habits [1]. While the definitions of “healthy” foods differ, generally, nutrition standards emphasising healthier foods would decrease the levels of sodium, sugar and unhealthy fats, while increasing the intake of fruits and vegetables, whole grains and lean protein [2]. On the other hand, unhealthy food consumption, such as consuming ultra-processed food, high calories and high-fat food, is a proven risk factor for a variety of chronic diseases that cause millions of deaths and disabilities each year globally [3,4,5,6,7,8]. Previous studies have showed that the prevalence of overweight and obesity among type-2 diabetes patients was 36.2% in Southwest Ethiopia; while 28.1% and 5.2% of overweight and obesity adults respectively in China had associations with factors such as food consumption, gender, income level, family history, physical inactivity and marital status [9,10,11]. Consequently, encouraging healthier food purchasing and consumption is critical for population health.

In recent years, around 75 percent of individuals have used various types of mobile interventions [12], and the mobile phone has become an essential medium of communication throughout the world [13]. Apps on mobile devices, such as applied on sensors and mobile apps, are becoming the centre of customers’ digital lives, and they are widely used. Many interventions are linked to the widespread usage of smartphones and apps nowadays. Apple’s iPhone, the first modern smartphone with the ability to add apps, was released just over ten years ago, in 2007 [14]. The smartphone, which is essentially a computer in your pocket, has become both personalised and easily accessible in recent years. As eating is an everyday activity, the prospect of altering behaviour using digital technologies in general is very enticing. Therefore, the use of technology apps intervention could be a promising approach towards altering people’s behaviour towards becoming a healthier consumer.

There has been an increase in the number of studies that have used technology interventions to alter people’s behaviour toward becoming healthier [15,16,17,18,19]. Technology-based behavioural health interventions involve the delivery of evidence-based practises via text messaging, apps, social media and multiple health components [20,21,22], through platforms such as computers, mobile phones or wearable sensors [23]. Due to rapidly growing evidence of their effectiveness and efficiency, technology-based behavioural health interventions are gaining traction as therapeutic resources, both as stand-alone technology or as multi component interventions [24,25,26,27,28,29,30,31,32,33]. Reflecting on this trend, there are reviews on the efficacy of stand-alone technology apps intervention or multicomponent intervention on healthy lifestyles [34,35,36,37,38,39,40,41], but still, the developing literature base on technology-based intervention has primarily focused on disease prevention [29,38,42,43], weight management [15,16,18,44,45] or lifestyle improvement outcomes [17,19,44,46], and in many ways has not yet explicated the important implications for a stand-alone technology intervention on healthy food purchases and healthy food consumption outcomes. Along with the relative lack of attention, specifically on healthy food purchasing and consumption, this review paper poses the following important research questions: Are technology app-based interventions effective enough to support healthy food purchase and consumption when compared with the traditional interventions such as face-to-face counselling, literature and group-based education? In which form are technology apps more effective and commonly used by adults? Is the effectiveness of stand-alone technology apps intervention comparative with the multi-component intervention? Which group of people have more access to technology apps? In order to fill in the knowledge gap, the general purpose of this review is to examine the efficacy of interventions that use technology-based apps to improve both healthy food purchasing and consumption in adults. Specific objectives were: (1) to determine the effectiveness of technology apps-based intervention to support healthy food purchase and consumption; (2) to identify the types of technology apps that are more effective and commonly used by adults; (3) to determine the potential of efficacy of stand-alone technology apps intervention on healthy food purchasing and consumption; and (4) to identify the target group of people who have more access to technology apps.

## 2. Materials and Methods

The study methods were registered in the international prospective register of systematic reviews (PROSPERO) protocol, with the registration number: CRD42020218742 [47]. We followed the Preferred Reporting Items for Systematic Reviews and Meta-Analyses (PRISMA) guidelines when conducting this systematic review and reporting its results [48].

### 2.1. Searching Strategy

The electronic databases searched were PubMed, Scopus, CINAHL, SPORTDiscuss and ACM Digital Library. These databases were searched for studies published in journals between 1 January 2006 and 31 December 2020, as it was considered unlikely that apps interventions were developed before 2006/2007, when smartphones were first introduced. The search was limited to English-language publications, and restricted to studies focusing on apps on food purchasing and food consumption in adults. Table 1 presents the search terms used in the present review.

### 2.2. Eligibility Criteria

Table 2 shows the domains, inclusion and exclusion criteria. The inclusion criteria were: (1) Publication years between 1 January 2006 and 31 December 2020; (2) original articles published in peer-reviewed journals; (3) studies in the English language; (4) studies from any country and any population; (5) targeted adults who are aged 18 and above; (6) quantitative studies with experimental or cross sectional study design that reported technology apps as an intervention group; and (7) studies must contain the food purchasing or food consumption as the outcome.

### 2.3. Study Selection

First, two investigators imported the potential articles from the databases into the Endnote Program X5 Version separately. Any duplicate publications were removed by Endnote Program X5. A manual check of the redundant publications was performed by the two investigators. Second, two investigators independently screened the titles and abstracts of those articles for suitability based on the search strategies mentioned above. Third, full-text articles were independently assessed based on the inclusion and exclusion criteria. Lastly, all articles that were agreed upon after discussing them with the third investigator were included.

### 2.4. Data Extraction

The following data were extracted by two investigators independently using a standardised data extraction template with the information such as the first author, year of publication, country, study name or app name, device, study design, sample characteristics, number of participants, mean age, percentage of females, grouping, intervention time frame, outcomes and measures and the finding of the studies. The results of data extraction were compared between the two investigators to ensure no errors.

### 2.5. Quality Assessment

We used Transparent Reporting of Evaluations with Nonrandomised Designs (TREND) to assess the quality of included studies [49]. This tool consists of 46 items that assessed components in non-randomised trials. Whenever the information provided was not enough to assist in making judgement for a certain item, we graded the item with a “0” meaning absence of component, and “1” was awarded to a particular item if information on a particular component was sufficiently described. Each article’s quality was graded as “Good” if the TREND score ≥37 or equivalence to 80%; or “Poor” if TREND score was <37. In this review, we would not exclude any studies based on their TREND score. The detail of scoring results is shown in Appendix A.

### 2.6. Data Analysis

All authors read, understood and synthesised the findings. Several discussions were conducted among the team to categorise findings into the following themes: “efficacy”; “technology apps”; “unhealthy foods”; and “healthy foods”. We follow the definition as below:Efficacy was defined as the ability to produce a desired or intended result [50].Technology apps were defined as applications for novel mobile consumer devices with a touchscreen, especially smartphones and tablet PCs [51]. In this review, the terms “app” is sued in the stricter sense with more on self-monitoring apps. Other technology interventions such as text messaging, social media (e.g., facebook), online coaching or telephone counseling were excluded in this review.Unhealthy foods were defined as those that were high in salt, sugar and saturated fats [2].Healthy foods were generally defined as the food meeting the nutrition standards promoting healthier foods that were low in sugar, salt and saturated fats, while promoting fruits and vegetables, whole grains and lean protein [2].

## 3. Results

### 3.1. Description of Included Studies

A total of 4175 articles were identified in the initial screening, as shown in Figure 1. After the removal of duplicate articles (*n* = 824), a total of 3351 articles were retrieved for the review of their title and abstract to determine if inclusion and exclusion criteria were met. Of the 3351 abstracts, 147 studies were identified that needed a further full-text review. After the careful evaluation of the 147 articles, 20 articles were included in this systematic review.

### 3.2. Characteristics of Included Studies

The characteristics of the included studies are summarised in Table 3. In general, most of the studies were conducted in the United States (*n* = 9, 45%) [24,25,26,30,32,52,53,54,55], followed by Australia (*n* = 2, 10%) [56,57] and Korea (*n* = 2, 10%) [28,33]. The majority of studies were from developed Western countries (*n* = 16, 80%) [19,24,25,26,27,29,30,31,32,52,53,54,55,56,57,58], followed by three Asian countries (Korea and India) [28,33,59] and one Western Asian country (Saudi Arabia) [60]. Most of the technology app devices were smartphones, and only two early studies used a personal digital assistant (PDA). The sample size ranged from 27 to 833, and the mean age of the participants ranged from 19 to 65 years. All studies involved a majority of female participants as compared to male participants. Overall, 14 out of 20 identified studies reported significant improvements in healthy food consumption behaviour and related healthy eating outcomes. On the other hand, only two studies reported intervention apps group as having significant improvements in overall household purchasing of healthy food choices.

### 3.3. Quality Assessment

The summary of the quality assessment is presented in Table 3, and the detail of scoring result was shown in Appendix A. The TREND score ranged from 28 to 39 (total mark is 46), or equivalent to 60.9% to 84.8%. Out of 20 included studies, four studies were marked as “good” quality and 16 studies remarked as “poor” quality. The most common absent component (none of studies reported these components) seen across included studies were “8c: Inclusion of aspects employed to help minimise potential bias induced due to non-randomisation (e.g., matching), 12b: Description of protocol deviations from study as planned, along with reasons”, 14c: Baseline comparisons of those lost to follow-up and those retained, overall and by study condition” and “17c: Inclusion of results from testing pre-specified causal pathways through which the intervention was intended to operate, if any”.

### 3.4. Healthy Food Purchasing and Its Measurement

There are two studies that specifically showed the household food purchases [29,54]. Eyles et al. [29] aimed to determine the effectiveness of a technology app to support people with cardiovascular disease to make low-salt food choices [29]. The primary outcome of the study was the salt content of household packaged food purchases during the four-week intervention (g/MJ). Secondary outcomes were the saturated fat content (g/MJ), energy content (kJ/kg) and expenditure (NZ$) of household food purchases. Results showed a significant reduction in mean household purchases of salt, but were not significant on the saturated fat content (g/MJ), energy content (kJ/kg) and expenditure (NZ$) of household food purchases, in the intervention app group as compared with the control group.

On the other hand, Palacios et al. [54] aimed to pilot test the effectiveness of a smartphone app that generates healthy grocery lists, on the indices of diet and weight. Results showed a significant improvement in the app group compared to the baseline, but no significant improvement compared to the traditional group (face-to-face counselling sessions) [54]. Although other included studies did not specifically measure the healthy food purchase, the measurement of healthy food choices may correlate with healthy food purchases. The summary of the healthy food purchase results is showed in Table 3.

### 3.5. Healthy Food Consumption and Its Measurement

The measurement of healthy food consumption varied across all studies. The measurement used included food choice (varies from type of food: vegetables, fruits, whole grains, sugar-sweetened beverage, unhealthy snacks, fatty food and alcohol consumption), dietary intakes (measured by calories, energy, total fat, carbohydrates, protein and fibre intake), nutrients intake (macronutrient: carbohydrate, fat and protein, or micronutrients: vitamins and minerals intake or fibre consumption), healthy eating index (HEI), healthy eating behaviour and healthy dietary behaviour. The instrument to measure food intake included self-reported or face-to-face interview using two-day food records, food frequency questionnaire (FFQ), 24-h dietary recalls or self-administrated questionnaire.

Some researchers collected food consumption data for other purposes of analysis. For example, some researchers used apps to monitor the dietary intake and assist in weight management [24,26,59]. Dodd et al. (2017) used apps to provide lifestyle advice to pregnant women [56]. Gill et al. (2019) aimed to determine the influence of an app to improve healthy eating to reduce modifiable risk factors for chronic disease [58]. Huberty et al. (2019) aimed to test the efficacy of a mobile app on stress, mindfulness and self-compassion in college students [55]. Other published papers aimed to test the efficacy of apps in changing healthy lifestyle among patients [28,33,53]. The summary of the healthy food consumption results is showed in Table 3.

### 3.6. Intervention Efficacy

A summary of intervention effects for the included healthy food purchase and healthy food consumption outcomes (macro and micronutrients, healthy food types intake, healthy eating index and healthy eating behaviour) are presented in Table 4. Overall, only two studies discussed the healthy food purchasing as a outcome [29,54]. Both studies showed no significant difference between apps and the control group for healthy food purchasing, except Eyles et al. (2017), which demonstrated the finding that intervention group chose lower salt content when making food purchases.

A slightly larger proportion of studies reported vegetables (15 out of 21; 71%), fruits (14 out of 21; 67%) and grains intake (6 out of 21; 29%) as healthy food consumption. Only two studies showed significant improvements in vegetables [27,52] and fruits [27] intake in favour of the app intervention group. However, Rodgers et al. (2015) also showed that participants with a BMI < 21 kg/m^2^ have lower consumption of fruit intake in the app intervention group. Two out of 12 studies showed improvement in lower fatty food consumption in the intervention group [19,25]. One out of six studies had findings of more grains intake in the intervention group [52]. For the unhealthy food choices such as added sugar, salt and unhealthy snacks, there is no significant difference between the intervention group, nor within intervention and control group, except one study that showed that the intervention app group has lower unhealthy snack consumption than that of the traditional group (face-to-face counselling session) [54]. Furthermore, 50% of the studies demonstrated a positive result i.e., app intervention improves healthy eating behaviour [31,32,58].

## 4. Discussion

This systematic review found modest evidence for the efficacy of app interventions to improve healthy food purchasing and food consumption. The majority of these studies found that significant between-group improvements in the apps intervention group versus the comparison group, which is the gold standard for evaluating the efficacy of health interventions [61]. Despite the limitations of the studies, the results of this review suggest that apps can be an important tool for improving healthy food purchasing and healthy food intake. The effectiveness of apps interventions can be explained in part by the advantages of mobile applications over other intervention delivery modes such as text messaging, websites, paper journals, email, face-to-face counselling and group sessions. Since many people lead busy lives, they may prefer applying easier access to health programs using smartphone or PDA apps anytime and anywhere, rather than spending time on looking for the paper journals, or making appointment for the face-to-face counselling or group sessions at particular times. Besides, the well-developed apps may provide the sufficient information and guidance, real-time self-monitoring, feedback, motivation, social support and incentives when they achieve the goals, in assisting them with health behavioural changes [62]. In this sense, technology apps are effective tools in this era where most people are looking for convenient, cost and resource savings interventions.

Among the included studies in this review, since 2006, when the integrated applications were launched, hand-held computers were utilised to improve and monitor the dietary intakes of the consumers [25,52]. Only until recently, smartphone applications were utilised in obesity treatment to achieve weight loss among smartphone users [24,26,57,60]; as well as in salt reduction movements, through buying low salt content food purchase behaviours [29]. Western countries were still the leading countries for the technology app penetration when compared to Asian countries. During the pre- and post-COVID-19 pandemic, smartphone applications were largely adopted among consumers due to their accessibility, trust, user friendliness, satisfaction and great performance, with almost half of consumers (45.68%) using the apps at least once every three days [63]. During the COVID-19 pandemic era, many countries were locked down, and the mandatory closure of gyms and outdoor recreation venues (such as parks, gardens, swimming pools, basketball courts and stadiums) may limit one’s movement in their own house, and consequently cause increasing levels of sedentary lifestyle among the population. Health-related issues have been in rather alarming increments, and technology apps may be an effective tool to engage people in healthy behaviours. Hence, the integrated technology applications in embracing both healthy food purchase and consumption were considered practical solutions to prevent health problems such as obesity and other non-communicable diseases, while promoting healthy food purchase and consumption at the same time, akin to fixing two major health concerns simultaneously. Technology-based apps are expected to be used by more people globally regardless of geographical locations in the very near future. In recognising this need, many governments around the globe, including in Malaysia, have subsided the provision of cheaper internet, or even smartphones, to enable people to remain connected, especially with regards to health-related issues.

Notwithstanding the potential of the apps, most of the intervention studies included in this review showed the significant findings only in selected food intakes, but not all, which means that the comparison group is still playing its role in health intervention. Technology apps are a motivational and self-monitoring tool, but it is highly dependent on the participant’s motivation and intention to use the app. Studies suggested that more intensive follow-ups with apps notifications may enhance the efficacy of the apps intervention [54,59]. To counter this, technology apps must provide the user-desired needs of the combination of credentialed knowledge, interactivity, personalisation, and individual feedback, in addition to providing information about positive food-related behaviour changes [64,65]. On the other hand, the insignificant findings of the between-group improvements of some of the studies also raises the question whether multi-component interventions have a greater impact than single-app interventions. To achieve long-term health behaviour improvement, several reviews of health behaviour change programmes suggest using multiple intervention techniques to achieve better outcomes [66,67,68,69]. As a result, it is possible that integrating apps into multi-component interventions yields better health outcomes than stand-alone app interventions; however, future studies will need to confirm this.

All of the included studies showed that younger aged persons formed the majority of participants in the intervention apps group, and the majority of participants were females. It demonstrated that age and gender affected smartphone usage, and also contributed to the significance in results. It is supported by research which reported that younger adults use their phones for longer periods of time, and their usage is primarily directed toward entertainment and social interactions via specialised apps [70]. Older participants use them less frequently, and primarily for information or as a traditional phone [70]. Furthermore, females use smartphones for longer periods than males, with an average of 166.78 min vs. 154.26 min per day, respectively [70]. In terms of purchasing behaviours, female purchasers opted to choose healthier food options such as vegetables and fruits, when compared to male purchasers [71]. Women from lower socioeconomic income groups, on the other hand, were considered as purchasing less healthy food choices that did not manage to meet the recommended dietary intakes for vegetables, milk and oils [72]. The individuals from the lower socioeconomic income group also possessed higher intention to consume oversized portions of unhealthy food, when compared to those from the higher socioeconomic group [73]. With this finding, technology app interventions should be more effective for younger adults and female users. Future designs for app development should be based on differences between the different groups of individuals.

Only two studies evaluate the effectiveness of apps on healthy food purchasing, showing improvements within the apps intervention group, in selected food categories [29,54]. However, two particular studies were directed at different health behaviour outcomes such as weight management [54] and lower salt intake for cardiovascular disease patients [29]; this is evidence that the existing technology apps clearly lack a focus on healthy food purchasing in improving general health. Given the importance of food purchasing behaviour as an important step in the food consumption process [74], as well as the potential effectiveness of mobile apps to support behaviour change, there is a need to further investigate the potential role of existing mobile apps.

### Strengths and Limitations

This review synthesises the existing literature and identifies any remaining gaps. It generates a comprehensive summary and discussion on the topic about the technology apps intervention on healthy food purchase and healthy food intake. The research characteristics, study designs, intervention durations, outcomes and findings are all covered in this review. Based on our analysis, a number of gaps in this field were identified, as highlighted in the discussion section above.

Another strength of this review is that it only identifies the efficacy of technology apps on food purchasing and food intake. It extracts the technology app intervention findings from multi-component interventions. Moreover, it draws on the food purchasing and food intake data from the studies which are not focused on food purchases or food consumption alone. For instance, studies focusing on weight loss, mindfulness meditation and lifestyle modification for a targeted group of patients were included in this review.

Potential biases may have affected the findings of this review, which is one of its limitations. First, limiting this review only to English language-based studies as one of the selection criteria may cause a selection bias. It introduces the risk of ignoring data that is not published in the English language, and may cause missing out on important cultural contexts which are tied to geographical aspects; this may therefore limit the review’s findings. However, a systematic review has demonstrated no evidence of systematic bias from the application of language restrictions on choosing the articles [75]. To assess the effect of language restrictions on systematic reviews in health sciences, further research is required. Secondly, publication bias could be present, which could be due to the lack of negative effects of the reported interventions in this review. Lastly, we found that the interventions mentioned in older studies from 2013 vary from those described in newer studies. Technology apps with older interventions may not be as effective as newer apps with advanced technology such as food photo recognition and personalised real-time feedback.

## 5. Conclusions

In conclusion, technology apps are an effective media, and are comparative with the traditional interventions for improving healthy food purchasing and nutrition-related outcomes among many diverse communities, especially among younger adults and female users. Smartphone apps are more effective, accessible and commonly used by adults, and can be used for stand-alone technology app intervention, which has the comparable positive effect with the multicomponent intervention. The strengths of technology apps are attributed to their nature of low cost, wide reachability and people having more time available to browse for online purchasing platforms, especially during the COVID-19 pandemic. However, the need for continued engagement and certain complicated app features are the key factors that can limit its effectiveness. Therefore, app development should be tailored with a different degree of healthy food choices. There is definitely more room for development and testing, particularly in this ever-vibrant app market, to establish its long-term efficacy.

## Figures and Tables

**Figure 1 foods-10-01861-f001:**
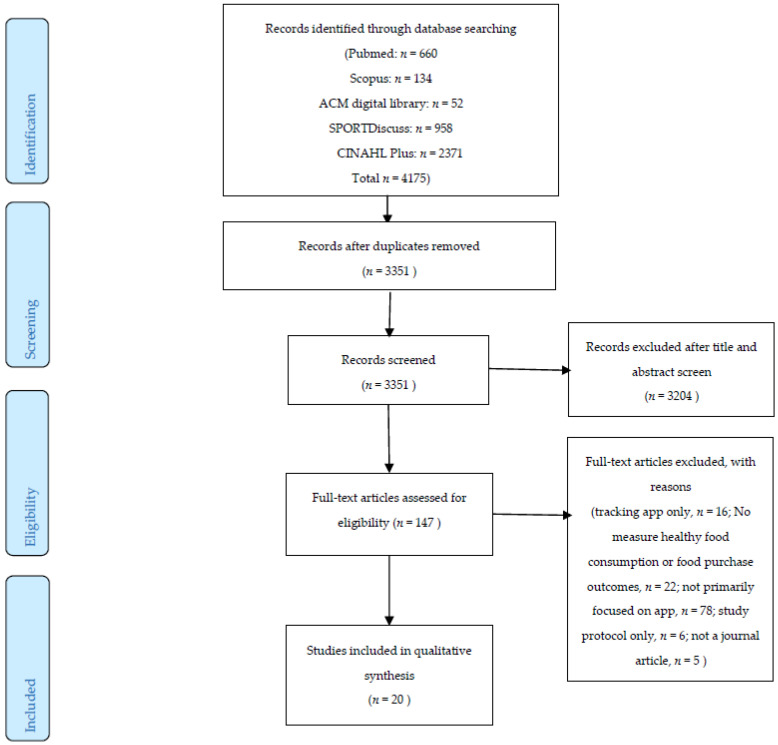
Preferred Reporting Items for Systematic Reviews and Meta-Analyses (PRIMSA) flow diagram of the literature screening process.

**Table 1 foods-10-01861-t001:** Search terms.

Search Category	Search Terms
Apps	Application* OR app* OR smartphone* OR “smart phone*” OR “cellular phone*” OR “mobile phone*” OR “mobile telephone*” OR tablet* OR “e-learning” OR “e-health” OR “iPad*” OR “mobile health” OR “social media”
Food purchasing and Food consumption	“food purchasing” OR “food purchase*” OR “food choice*” OR “food planning” OR “food shopping” OR “food consumption*” OR “food intake*” OR “dietary intake*” OR “healthy eating” OR “eating behaviour*” OR “healthy food” OR “food behaviour*” OR “food value*” OR “food diary” OR “food diaries” OR “nutrition assessment*” OR “diet record*” OR “diet survey*” OR “energy intake*” OR “nutrition survey*” OR “dietary assessment*”
Intervention	Intervention* OR program* OR programme* OR “health promotion*” OR trial* OR effectiveness

**Table 2 foods-10-01861-t002:** Inclusion and exclusion criteria.

Domain	Inclusion Criteria	Exclusion Criteria
1. Publication year	Studies published between 1 January 2006 and 31 December 2020.	Studies published before or after the inclusion dates.
2. Publication type	Original studies published in peer-reviewed journals only.	Letters, commentaries, conference proceedings, reviews, narrative articles or other materials that was not a peer-reviewed primarily study.
3. Language	Studies limited to English-language publications.	Studies were not published in English.
4. Targeted population	Any population.	No restriction on population.
5. Targeted group	Adults who are aged 18 or above.	Children or adolescents below 18 years old.
6. Research design	Quantitative studies involved experimental and cross-sectional study design.	Qualitative study or any non-experimental study designs were utilised (e.g., protocol, studies reporting prevalence or trend data, feasibility studies, measurement studies or theoretical papers).
7. Study scope/variables	(1) Technology apps [e.g., smartphone or personal digital assistant (PDA)] in an intervention to improve healthy food purchasing and consumption for prevention. (2) Interventions could be stand-alone interventions using an app only, or multi-component interventions including an app as one of several intervention components (e.g., physical education, face-to-face counselling) with the condition that individual apps record was provided. (3) All types and units of measurements for the healthy food purchasing and consumption outcomes were acceptable (e.g., food group, self-report, servings, calories, kilograms, nutrition assessment). (4) reported data regarding efficacy for behaviour change (e.g., change in healthy food groups). (5) Interventions covering aspects of large-scale management of food (retail, restaurants, public preparation and consumption of food in school kitchens, hospitals, etc.).	(1) tracking app only but not focused on apps intervention on food purchasing or food consumption. (2) Other technology intervention than apps intervention (e.g., text messaging, apps, social media, telephone counselling or online coaching). (3) No measure of healthy food purchasing or food consumption outcomes.

**Table 3 foods-10-01861-t003:** Characteristics and outcomes of studies evaluating efficacy of app-based intervention for supporting healthy food purchasing and consumption.

First Author (Year), Country	Study Name/Apps Name	Device	Study Design	Sample Characteristics, Mean Age	Female (%)	Grouping	Intervention Time Frame	Outcomes and Measures	Findings	Quality
Allen (2013), USA [24]	SLIM (Smart coach for LIfestyleManagement) study	smartphone	RCT	68 obese adults	78%	(1) intensive counseling intervention, (2) intensive counseling + smartphone intervention, (3) a less intensive counseling + smartphone intervention, and (4) smartphone intervention only	baseline and 6-month	self-reported dietary intake	Not significant.	Poor
Alnasser (2019), Saudi Arabia [60]	*Twazon* app.	smartphone	pre-post single-arm pilot study	40 overweight adult; engaged: *n* = 26, age = 31 years, Unengaged: *n* = 14, age = 31 years	100%	engaged users (65%) and unengaged users (35%)	baseline, 2- and 4-months	Dietary intake	The daily energy consumption was decreased by >600 calories in the engaged users group compare with the unengaged group.	Poor
Atienza (2008), USA [52]	NR	PDA	RCT	27 healthy mid-life and older adults (≥50 years); AG: *n* = 16, age = 63 years; CG: *n* = 11, age = 58 years	AG: 69% CG: 70%	PDA program vs. control	baseline and 8 weeks	vegetable and whole-grain intake	Intervention participants reported significantly greater increases in vegetable servings and dietary fibre from grains.	Poor
Banerjee (2020), India [59]	S Health^®^, Calorie Counter—MyFitnessPal^®^, and Calorie Counter	smartphone	prospective controlled trial	58 healthy young adults (18–45 years); AP: *n* = 30; CG: *n* = 28	AG: 63%; CG: 46%	apps group vs. control	baseline and 8 weeks	Food consumption	Not significant.	Poor
Brindal (2019), Australia [57]	MotiMate	smartphone	RCT	88 healthly adults; AG: *n* = 45, age = 45 years; CG: *n* = 43, age = 46 years	AG:75%; CG: 69%	intervention app (MotiMate) vs. control app	baseline, 4, 8, 12 and 24 weeks	healthy eating	Not significant.	Poor
Dodd (2017), Australia [56]	SNAPP trial	smartphone	RCT	162 healthy pregnant women; AG: *n* = 77, age = 31 years; CG: *n* = 85, age = 31 years	100%	Lifestyle Advice + Smartphone App vs. Lifestyle Advice Only	baseline, 28 and 36 weeks	healthy eating index (HEI)	Not significant.	Good
Eyles (2017), New Zealand [29]	SaltSwitch	smartphone	RCT	66 adults with diagnosed cardiovascular disease; AG: *n* = 33, age = 64 years; CG: *n* = 33, age = 65 years	AG:9%; CG: 24%	SaltSwitch app vs. control group (usual care).	baseline and 4 weeks	(1) salt content of household packaged food purchases (2) saturated fat content (g/MJ), energy content (kJ/kg) and expenditure (NZ$) of household food purchases	A significant reduction in mean household purchases of salt was observed. Not significant for the second outcome.	Good
Gill (2019), Canada [58]	HealtheSteps™	smartphone	RCT	118 adults at risk or diagnosed with a chronic disease; AG: *n*= 59, age = 57 years; CG: *n* = 59, age = 59 years	AG:76%; CG: 81%	HealtheSteps™ smartphone app and Healthe-Steps™ website vs. wait-list control	baseline and 18 months.	self-reported eating habits	Improved their overall healthful eating	Good
Glanz (2006), USA [25]	NR	PDA	Intervention pilot test	33 healthy women, 64 years	100%	PDA diet-monitoring system	baseline and 1 month	food choice and dietary intakes	Reported total fat intake and percent energy from fat decreased significantly.	Poor
Huberty (2019). USA [55]	Calm	smartphone	RCT	88 healthy adult; AG: *n* = 41, age = 20 years; CG: *n* = 47, age = 22 years	AG:41%; CG: 49%	Calm app vs. wait-list control	baseline, 8 and 12 weeks	alcohol consumption and healthy eating (fruit and vegetable consumption)	Not significant.	Poor
Inauen (2017), USA [30]	NR	smartphone	RCT	140 healthy adult; AG: *n* = 70, age 27.5 years; CG: *n* = 70. Age = 27.5 years	75.5%	Whatsapp support group (1. eating more fruit and vegetables 2. eating fewer unhealthy snacks) vs. control	baseline, 1- and 2-months	Self-reported healthy eating (fruits, vegetables and unhealthy snacks)	Intervention group showed a gradual increase in healthy eating over time, ate more fruits and vegetables, and less unhealthy snacks compare to the control group on Day 10. However, it is not significant at the follow ups.	Poor
Jarvela (2018), Finland [31]	NR	smartphone	RCT	219 healthy adult; face to face group: *n* = 70, age = 50 years; AG: *n* = 78, age = 49; CG: *n* = 71, age = 49 years	(1) Face-to-face: 87% (2) AG: 85% 3) CG: 82%	(1) Face-to-face (2) mobile app (3) control	baseline, 10 and 36 weeks	eating behaviour	App group showed beneficial effects on reported eating behaviour.	Poor
Lee (2019), Korea [33]	NR	smartphone	RCT pilot test	65 adult who diagnosis of colorectal polyps; AG: *n* = 32, age = 49 years; CG: *n* = 33, age 21 years	AG:34%; CG: 46%	intervention app vs. control (traditional mail)	baseline and 3 month	changes in dietary intake, such as that of vegetables, fruits, and fatty food.	Not significant.	Poor
McCarroll (2015), USA [53]	LoseIt!	smartphone	Prospective intervention	50 adult women cancer survivors, age = 58 years	100%	web- or mobile-based apps	baseline and 4 weeks	macronutrient (carbohydrate, fat and protein) and fibre consumption	Not significant.	Poor
Palacios (2018), USA [54]	MyNutriCart	smartphone	pilot randomised trial	51 overweight and obese adult; AG: *n* = 24, age = 34 years; TG: *n* = 27, age = 37 years	AG:92%; TG: 89%	intervention app vs. face-to-face counseling session	baseline and 8 weeks.	healthy food choice and dietary behaviour	“MyNutriCart” app use led to significant improvements in food-related behaviours compared to baseline, with no significant differences when compared to the traditional group.	Poor
Park (2016), Korea [28]	Strong bone, Fit body (SbFb)	smartphone	RCT	82 young adult women with low bone mass; AG: *n* = 28, age = 24 years; Group education: *n* = 32, age = 25 years; CG: *n* = 22, age = 23 years	100%	(1) apps (2) group education (3) control	baseline and 20 weeks	nutrient intake	calcium intake is higher in app and group education than control group.	Poor
Recio-Rodriguez (2018), Spain [19]	EVIDENT II study	smartphone	RCT	833 healthy adult; AG: *n* = 415, age = 51 years; CG: *n* = 418, age = 52 years	AG:60%; CG: 64%	intervention: counseling + application group; control: counseling group	baseline and 12-month	Macro and Micronutrients intake	The app group reported a higher percentage intake of carbohydrates, and lower percentage intakes of fats and saturated fats	Good
Rodgers (2015), France [27]	NR	smartphone	Intervention only	40 healthy female adults, age = 19 years	100%	intervention: app (food journal + messages)	baseline and 3 weeks	fruit, vegetable, and sugar-sweetened beverage consumption.	Among participants with body mass index (BMI) ≥25, fruit and vegetable consumption increased with time. Among participants with BMI <21, consumption of fruit decreased, whereas the consumption of vegetables remained stable. No effects were found for sugar-sweetened beverage consumption.	Poor
Sarcona (2017), USA [32]	NR	smartphone	cross-sectional study	401 university students	73%	Users and Nonusers of Mobile Health Apps	NA	healthy eating behaviour	App users were found to have more positive eating behaviours than nonusers, and the impact of using more than one type of mobile-based health app significantly improved eating behaviour.	Poor
Turner (2013), USA [26]	Fat Secret’s Calorie Counter, My Fitness Pal, and Lose it	smartphone	RCT	78 overweight and obese adult; AG: *n* = 37, age = 41 years; website: *n* = 24, age = 45 years; paper journal: *n* = 17, age = 47 years;	AG: 70%; website: 87%; paper journal: 76%;	(1) mobile app, (2) website, and (3) paper journal	baseline and 6 months	dietary intake (energy intake, fat, added sugar, fruit, vegetables) and eating behaviour	App users consumed less energy than paper journal users. No significant difference on the dietary intake and eating behaviour.	Poor

PDA: Hand-Held Computer (personal digital assistant); RCT: randomised controlled trial; AG: App group; CG: control group; TG: traditional group.

**Table 4 foods-10-01861-t004:** Summary of intervention effects on healthy food purchasing and food consumption outcomes.

Study	Healthy Food Purchasing	Healthy Food Consumption													
		Energy Intake	Carbohydrate	Protein	Fat	Micronutrients	Grains	Vegetables	Fruits	Fibre	Added Sugar	Salt	Unhealthy Snack	Healthy Eating Index	Healthy Eating Behaviour
Allen (2013) [10]		0			0			0	0			0			
Alnasser (2019) [39]		+ (b)		0	0			+ (b),	+ (b),		+ (w)				+ (w)
Atienza (2008) [30]							+ (b)	+ (b)		+ (b)					
Banerjee (2020) [38]					0						0		0		
Brindal (2019) [35]				0			0	0	0					0	
Dodd (2017) [34]			0	0	0	0	0	0	0		0	0		0	
Eyles (2017) [15]	+ (b) (salt content); 0 (saturated fat, energy content and expenditure)														
Gill (2019) [37]					0			0	0		0				+ (b)
Glanz (2006) [11]		+ (b)			+ (b)		0	0	0	0					
Huberty (2019) [33]								0	0						
Inauen (2017) [16]								+ (b) (at day 10); 0 (at 2 months)	+ (b) (at day 10); 0 (at 2 months)				+ (b) (at day 10); 0 (at 2 months)	+ (b) (at day 10); 0 (at 2 months)	
Jarvela (2018) [17]															+ (b)
Lee (2019) [19]					+ (w)			+ (w)	+ (w)						
McCarroll (2015) [31]		0	0	0	0					0					
Palacios (2018) [32]	+ (w)		0	0	0	0	0	0	0		0		+ (b)		
Park (2016) [14]		0	0	0	0	+ (b)									0
Recio-Rodriguez (2018) [36]		+ (w)	+ (b), + (w)	0	+ (b), + (w)	0	0	0	0	0					0
Rodgers (2015), (participants with BMI ≥25) [13]								+ (b)	+ (b)		0				
Rodgers (2015) (participants with BMI <21) [13]								0	− (b)		0				
Sarcona (2017) [18]															+ (b)
Turner (2013) [12]		+ (b)			0			0	0		0				0

+ (b): between-group significant improvements in favour of app intervention group, − (b): between-group significant improvements in favour of non-app control group, + (w): within-group significant improvement in favour of app intervention group, 0: no significant change.

## Data Availability

Data from this study is contained within this article and Appendix A.

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
