# Peer review of "Effectiveness of Integrated Technology Apps for Supporting Healthy Food Purchasing and Consumption: A Systematic Review"

_foods, 2021, doi:10.3390/foods10081861_

Round 1

Reviewer 1 Report

Aims: It would be easier to follow the structure of the paper if the authors would create a list from the aims. Additionally, at the end of the paper, these aims should be addressed one-by-one.

L128: “criteria of the criteria”

L169: do not capitalize Unhealthy here.

The authors aim to answer the following research questions:

  • Are technology apps-based intervention effective to support healthy food purchase and consumption when compared with the traditional interventions such as face to face counselling, literatures, and group-based education?
  • Is the effectiveness of stand-alone technology apps intervention comparative with the multicomponent intervention?
  • In which form are technology apps more effective and commonly used by the adults?
  • Which group of people are more accessible to technology apps?

Although the answers are present in the paper, it is important to restructure the discussion in a way which helps readers finding these answers.

Author Response

Manuscript No.: Foods- 1280445

Dear Reviewer 1:

General comment:

Point 1: Aims: It would be easier to follow the structure of the paper if the authors would create a list from the aims. Additionally, at the end of the paper, these aims should be addressed one-by-one.

Response 1: Thank you for your comment. We have now added the specific objectives in the introduction in page 2 line 92-98, to read as:

Specific objectives were: 1) to determine the effectiveness of technology apps-based intervention to support healthy food purchase and consumption; 2) to identify the types of technology apps that are more effective and commonly used by the adults, 3) to determine the potential of efficacy of stand-alone technology apps intervention on healthy food purchasing and consumption, 4) to identify the target group of people who are more accessible to technology apps.

We also added few sentences in conclusion part to address the aims one by one in page 16 line 366 – 373, to read as:

In conclusion, technology apps are an effective media and are comparative with the traditional interventions for improving healthy food purchasing and nutrition-related outcomes among many diverse communities especially among younger adults and female users. Smartphone apps are more effective, accessible and commonly used by the adults and it can be a stand-alone technology apps intervention which has the comparable positive effect with the multicomponent intervention

Point 2: L128: “criteria of the criteria”

Response 2: Thank you for your comment. We have now deleted “of the criteria” in page 3 line 116, to read as:

            Table 2 shows the domains, inclusion and exclusion criteria.

Point 3: L169: do not capitalize Unhealthy here.

Response 3: Thank you for your comment. We have now amended the “Unhealthy” to “unhealthy” in page 5 line 153.  

Point 4: The authors aim to answer the following research questions:

Are technology apps-based intervention effective to support healthy food purchase and consumption when compared with the traditional interventions such as face to face counselling, literatures, and group-based education?

Is the effectiveness of stand-alone technology apps intervention comparative with the multicomponent intervention?

In which form are technology apps more effective and commonly used by the adults?

Which group of people are more accessible to technology apps?

Although the answers are present in the paper, it is important to restructure the discussion in a way which helps readers finding these answers.

Response 4: Thank you for your comment. We have now restructured the research questions in page 2 line 84-90, to read as:

this review paper poses the following important research questions: Are technology apps-based intervention effective to support healthy food purchase and consumption when compared with the traditional interventions such as face to face counselling, literatures and group-based education? In which form are technology apps more effective and commonly used by the adults? Is the effectiveness of stand-alone technology apps intervention comparative with the multi-component intervention? Which group of people are more accessible to technology apps?

Reviewer 2 Report

The comments have been correctly addressed

Author Response

Dear reviewer:

     We appreciate your time and effort in giving comments to improve the quality of this manuscript. Thank you. 

This manuscript is a resubmission of an earlier submission. The following is a list of the peer review reports and author responses from that submission.

Round 1

Reviewer 1 Report

This article addresses the important and current issue of the effectiveness of various technological applications used to purchase and consume healthy foods. 
The topic is not new, but the research done should be considered a contribution of the authors. 
The content of the work concerns the topic given in the title of the work. The work is fairly consistent and you can see a smooth transition from one point to another. 
The authors point out some limitations that create opportunities for future research. In the article, the authors referred to the latest literature. However, I do note some shortcomings that would be worth considering. 
In the structure of the article, I miss an approximation of the main technological applications that are currently functioning and are the most popular. 
It is worth considering inserting a section: "Literature review", in which you could introduce the technological applications used, as well as the concept of "healthy food". The authors included their understanding of these concepts primarily in lines 120-132 of the article. 
The article does not indicate hypotheses or even research questions or specific goals. 
It is not entirely understandable for me to focus on 20 articles in the final analysis, as well as to select the databases from which the articles were downloaded. The selection (elimination) of these articles was not very precisely explained. Of course, the authors' contribution to the review of given sources should be emphasized. However, I expected broader and clearer practical implications.
 The conclusions are imprecise and too general. A small correction of the text in terms of language is required, eg in line 59 instead of "primary" it should be "primarily" or in some places prepositions are missing (e.g. the) or unnecessary

Author Response

Dear Reviewer 1:

General comment:

This article addresses the important and current issue of the effectiveness of various technological applications used to purchase and consume healthy foods.

The topic is not new, but the research done should be considered a contribution of the authors.

The content of the work concerns the topic given in the title of the work. The work is fairly consistent and you can see a smooth transition from one point to another.

The authors point out some limitations that create opportunities for future research. In the article, the authors referred to the latest literature. However, I do note some shortcomings that would be worth considering.

We would like to thank you for your valuable comments and suggestions to improve the quality of this manuscript. We have revised this manuscript accordingly and responded to the reviewer’s comments point by point as following: 

Point 1: In the structure of the article, I miss an approximation of the main technological applications that are currently functioning and are the most popular.

It is worth considering inserting a section: "Literature review", in which you could introduce the technological applications used, as well as the concept of "healthy food". The authors included their understanding of these concepts primarily in lines 120-132 of the article.

Response 1: Thank you for your insightful comment. We have now added one paragraph to explain “healthy food” in page 1 paragraph 1, to read as:

Healthy food purchase and consumption are the trending topics among consumers nowadays. Increase in healthy food purchasing patterns may motivate consumers to practice healthier food consumption habit [1]. While the definitions of “healthy” foods differ, generally, nutrition standards emphasizing healthier foods would decrease the levels of sodium, sugar and unhealthy fats while increasing the intake of fruits and vegetables, whole grains, and lean protein [2]. On the other hand, unhealthy food consumption, such as consuming ultra-processed food, high calories and high fat food are a proven risk factor for a variety of chronic diseases that cause millions of deaths and disabilities each year globally. Previous studies have showed that the prevalence of overweight and obesity among type-2 diabetes patients was 36.2% in Southwest Ethiopia; 28.1% and 5.2% of overweight and obesity adults respectively in China had associations with factors such as food consumption, gender, income level, family history, physical inactivity, and marital status [3-5]. Consequently, encouraging healthier food purchasing and consumption is critical for population health.

We have also added one paragraph to introduce technology apps in page 2 paragraph 2, to read as:

In recent years, around 75 percent of individuals have used various types of mobile interventions [6], and the mobile phone has become a very essential medium of communication throughout the world [7]. Apps on the mobile devices such as applied on sensors and mobile apps, are becoming the centre of customers' digital lives, and they are widely used. Many interventions are linked to the widespread usage of smartphones and apps nowadays. Apple's iPhone, the first modern smartphone with the ability to add apps, was released just over ten years ago, in 2007. The smartphone, which is essentially a computer in your pocket, has become both personalised and easily accessible in recent years. As eating is an everyday activity, the prospect of altering behaviour using digital technologies in general is very enticing. Therefore, the use of technology apps intervention could be a promising approach towards altering people's behaviour toward becoming a more healthier consumer.

Point 2: The article does not indicate hypotheses or even research questions or specific goals.

Response 2: Thank you for your nice comment. We have now added research questions in page 2 line 72 – 82, to read as:

Along with the relative lack of attention specifically on healthy food purchasing and consumption, this review paper poses the following important research questions: Are technology apps-based intervention effective to support healthy food purchase and consumption when compared with the traditional interventions such as face to face counselling, literatures and group-based education? Is the effectiveness of stand-alone technology apps intervention comparative with the multicomponent intervention? In which for mare technology apps more effective and commonly used by the adults? Which group of people are more accessible to technology apps? In order to fill in the knowledge gap, the general purpose of this review is to examine the efficacy of interventions that use technology-based apps to improve both healthy food purchasing and consumption in adults. 

Point 3: It is not entirely understandable for me to focus on 20 articles in the final analysis, as well as to select the databases from which the articles were downloaded. The selection (elimination) of these articles was not very precisely explained. Of course, the authors' contribution to the review of given sources should be emphasized. However, I expected broader and clearer practical implications.

Response 3: Thank you for your comment. For this review, we strictly followed the Preferred Reporting Items for Systematic Reviews and Meta-Analyses (PRISMA) guidelines when conducting this systematic review and reporting its results (26). The final 20 articles selected were after the process of screening as specified by the PRISMA guideline. In this review, there were 5 electronic databases selected. They were PubMed, Scopus, CINAHL, SPORTDiscuss and ACM Digital Library. PubMed, Scopus, CINAHL and SPORTDiscuss are the leading electronic database of references that contain peer-reviewed journals in top-level subject fields, namely life sciences, social sciences, physical sciences and health sciences. Nevertheless, as this review topic is about technology apps, we included ACM Digital Library as it is the top database to cover the computing and information technology related articles. We believed that these 5 databases were the most relevant databases to this study and can help us to get the most eligible articles. About the selection (elimination) process of the study, we strictly followed the inclusion and exclusion criteria as listed in Table 2. All search for articles was performed by 2 investigators independently. All articles that were agreed upon by these 2 investigators after discussing them with a third investigator were included.

Reference:

  1. Moher, D.; Shamseer, L.; Clarke, M.; Ghersi, D.; Liberati, A.; Petticrew, M.; Shekelle, P.; Stewart, L.A. Preferred reporting items for systematic review and meta-analysis protocols (PRISMA-P) 2015 statement. Systematic reviews 2015, 4, 1-9.

Point 4: The conclusions are imprecise and too general. A small correction of the text in terms of language is required, eg in line 59 instead of "primary" it should be "primarily" or in some places prepositions are missing (e.g. the) or unnecessary

Response 4: Thank you for your comment. We have now amended the "primary" to "primarily" in page 2 line 69. Furthermore, the conclusion has been revised in page 16 line 362-371, to read as:

In conclusion, technology apps are an effective media and are comparative with the traditional interventions for improving healthy food purchasing and nutrition-related outcomes among many diverse communities especially among younger adults and female users. The strength of technology apps are attributed to its nature of low cost, wide reachability, and people have more time available to browse for online purchasing platforms especially during thisCOVID-19 pandemic. However, the need for continued engagement and certain complicated app features are the key factors that can limit its effectiveness. Therefore, apps development should be tailored with different degree of healthy food choices. There are definitely more room for development and testing, particularly in this ever vibrant app market, to establish its long-term efficacy.

Reviewer 2 Report

Introduction: This part is not a real introduction as an introduction should help the readers to get familiar with the topic and the materials the paper will discuss. The whole section should be rewritten. Please indicate the short history of such apps, the reason we need these and what benefits can we get if we use them.

Materials and methods section is well-written, easy to follow and clear.

Results section is technical, the obtained information is well presented. The tables summarize the results well.

The discussion section supports the findings presented in the results section.

Conclusions are well-made.

Minor comment:

L20: healthy

Author Response

Dear Reviewer 2:

We would like to thank you for your valuable comments and suggestions to improve the quality of this manuscript. We have revised this manuscript accordingly and responded to the reviewer’s comments point by point as following: 

Point 1: Introduction: This part is not a real introduction as an introduction should help the readers to get familiar with the topic and the materials the paper will discuss. The whole section should be rewritten. Please indicate the short history of such apps, the reason we need these and what benefits can we get if we use them.

Response 1: Thank you for your insightful comment. We agree with you that the previous introduction was not pleasant to the reader familiar with the topic. We have now amended the introduction thoroughly in pages 1-2. We have added one paragraph in paragraph one to introduce the “healthy food”. We explained the definition and the important of “healthy food”, the association between unhealthy food and diseases, and the relationship between healthy food purchase and consumption. Furthermore, we have added one paragraph in paragraph two to introduce the technology apps. We have included the short history of apps and the reasons for why it is potential to influence healthy food purchasing and consumption behaviour. We hope the edited version may ease the reader familiar with the topic.

Point 2: Materials and methods section is well-written, easy to follow and clear

Response 2: Thank you so much your comment. It means a lot to us.

Point 3: Results section is technical, the obtained information is well presented. The tables summarize the results well.

Response 3: Thank you so much your comment.

Point 4: The discussion section supports the findings presented in the results section.

Response 4: Thank you so much your comment.

Point 5: Conclusions are well-made.

Response 5: Thank you so much your comment.

Point 6: Minor comment: L20: healthy

Response 6: Thank you for your comment. We have now amended the “health” to “healthy” in line 20 page 1.

Reviewer 3 Report

The article is relevant in the current scientific debate, and the research has been well conducted. However, the background information and the results should be presented in a clearer and more extensive way.

Introduction:

  1. Why refer to female? I think that household purchasers can be identified in a genderless way
  2. The background for this research should be presented: why apps can improve diets? For example they can fill the attitude-behavior gap, the construal level between short term and long term consequences of eating, they can cope with a unhealthy food environment, etc. All the mechanisms that apps can overcome should be presented here considering past literature. Also the contribution that this article provides to existing literature should be presented.

Materials and Methods:

3. The table 2 is very confusing. I suggest to shrink it in a more concise way, and delete the column rationale. The rationales behind each choice can be explained in a new paragraph.

4. Paragraphs 3.1 3.2 and 3.3 are still part of the methods section

Results: this section should be reorganized. So far no result is clear from the review.

5: the results should begin with a categorization of the analysed studies. This category should be at first designed and then presented. Subsequently, each category should be discussed in depth

6: Table 3 should be at the end of the paper

Discussion

7: Text from line 1 to 22 of this section should be part of a results section

8: the discussion of the topic in light of the covid restrictions should consider the reduced digital divide of the population of all countries and the reduced exercise of the consumers.

9: Are there any geographical patterns of behavior? Or the results are homogenous across countries?

Conclusions

10: The conclusions should summarize the whole study and wrap up the story. The existing sections is just a final comment on the article.

Author Response

Dear Reviewer 3:

We would like to thank you for your valuable comments and suggestions to improve the quality of this manuscript. We have revised this manuscript accordingly and responded to the reviewer’s comments point by point as following:

General comment:

The article is relevant in the current scientific debate, and the research has been well conducted. However, the background information and the results should be presented in a clearer and more extensive way.

Introduction:

Point 1: Why refer to female? I think that household purchasers can be identified in a genderless way

Response 1: Thank you for your comment. We agree with you that we should not discuss this specific factor that may influence the usefulness of technology apps in promoting healthy food purchasing and consumption. However, based on our review, we found that gender does play some role. Therefore, we will maintain the gender factor in the discussion section. We have now deleted the paragraph on gender and revised thoroughly the introduction in pages 1-2.

Point 2: The background for this research should be presented: why apps can improve diets? For example they can fill the attitude-behavior gap, the construal level between short term and long term consequences of eating, they can cope with a unhealthy food environment, etc. All the mechanisms that apps can overcome should be presented here considering past literature. Also the contribution that this article provides to existing literature should be presented.

Response 2: Thank you for your insightful comment. We agree with you that the previous introduction was not entirely clear to the reader unfamiliar with the topic. We have now amended the introduction thoroughly in pages 1-2. We have added a sentence into the first paragraph to introduce “healthy food”. We have now explained the definition and the importance of “healthy food”, the association between unhealthy food and diseases, and the relationship between healthy food purchase and consumption. Furthermore, we have also added a sentences into paragraph two to introduce the technology apps. We have included the short history of apps and the reasons for why it has the potential to influence healthy food purchasing and consumption behaviours. We hope the edited version may ease the understanding of readers unfamiliar with the topic.

Materials and Methods:

Point 3: The table 2 is very confusing. I suggest to shrink it in a more concise way, and delete the column rationale. The rationales behind each choice can be explained in a new paragraph.

Response 3: Thank you for your comment. We have now amended the Table 2 content. We have also deleted the rationales column in Table 2 and they are now explained separately in the main text. For example, the rationale to determine the publication years between 01 January 2006 and 31 December 2020 is explained in page 2 line 91-93, to read as:

These databases were searched for studies published in journals between 01 January 2006 and 31 December 2020 as it was considered unlikely that apps interventions were developed before 2006/2007 when smartphones were first introduced.

The rationale of restricting the studies published in English language was discussed in 4.1. Strengths and Limitations section in page 15 line 349 – 354, to read as:

First, limiting this review only to English language-based studies as one of the selection criteria may cause a selection bias. It introduces the risk of ignoring data that is not published in the English language and may cause missing out on important cultural contexts which is tied to geographical aspects. It may therefore limit the review’s findings. However, a systematic review has demonstrated that no evidence of systematic bias from the application of language restrictions on the chosen articles [53].

Point 4: Paragraphs 3.1 3.2 and 3.3 are still part of the methods section

Response 3: Thank you for your comments. However, we have reported that “We followed the Preferred Reporting Items for Systematic Reviews and Meta-Analyses (PRISMA) guidelines when conducting this systematic review and reporting its results [26].” In page 2 line 86-88. We are therefore restricted by the PRISMA statement for reporting. Therefore, we will maintain subsection 3.1,3.2 and 3.3 as it is in this manuscript.

Reference:

  1. Moher, D.; Shamseer, L.; Clarke, M.; Ghersi, D.; Liberati, A.; Petticrew, M.; Shekelle, P.; Stewart, L.A. Preferred reporting items for systematic review and meta-analysis protocols (PRISMA-P) 2015 statement. Systematic reviews 2015, 4, 1-9.

Results: this section should be reorganized. So far no result is clear from the review.

Point 5: the results should begin with a categorization of the analysed studies. This category should be at first designed and then presented. Subsequently, each category should be discussed in depth

Response 5: Thank you for your comments. We agree with you that the content of Tables 3 and 4 were disorganized. We have rearranged the order of studies according to author’s name. We hope that this arrangement will ease the readership.

Point 6: Table 3 should be at the end of the paper

Response 6: Thank you for your suggestion. We have reshuffled the arrangement of texts and tables in this manuscript as suggested. 

Discussion

Point 7: Text from line 1 to 22 of this section should be part of a results section

Response 7: Thank you for your comment. We do agree that the result should not presented in the discussion section. Therefore, we have now removed texts from lines 2 to 6 of this section and inserted it in the result subsection 3.2 in page 6 lines 170-174, to read as:

Overall, 14 out of 21 identified studies reported significant improvements in healthy food consumption behaviour and related healthy eating outcomes. On the other hand, only two studies reported intervention apps group as having significant improvements in overall household purchasing of healthy food choice.

Point 8: the discussion of the topic in light of the covid restrictions should consider the reduced digital divide of the population of all countries and the reduced exercise of the consumers.

Response 8: Thank you for your comment. We have now added the points suggested you and revised the paragraph in page 14, lines 272-289, to read as:

During the pre- and post- covid-19 pandemic, smartphone applications were largely adopted among consumers due to its accessibility, trust, user friendliness, satisfaction, and great performance with almost half of the consumers (45.68%) using the apps at least one time every three days [42]. During the COVID-19 pandemic era, many countries were locked down, and the mandatory closure of gyms and outdoor recreation venues (such as parks, gardens, swimming pools, basketball courts and stadiums) may limit one's movement in their own house, and consequently causing increasing levels of sedentary lifestyle among the population. Health related-issues has been reported in a rather alarming increments and technology apps may be an effective tool to engage people in healthy behaviours. Hence, the integrated technology applications in embracing both healthy food purchase and consumption were considered practical solutions to prevent health problems such as overweight, obesity and other non-communicable diseases, and yet promoting healthy food purchase and consumption at the same time, akin to fixing two major health concerns at the same time. The technology-based apps is expected to be used by more people globally regardless of geographical locations in the very near future. In recognising this need, many governments around the globe including in Malaysia has subsided the provision of cheaper internet or even smartphones to enable people to remain connected especially in health-related issues.

Point 9: Are there any geographical patterns of behavior? Or the results are homogenous across countries?

Response 9: Thank you for your comment. We have now added the points suggested you and revised the paragraph in page 14, lines 271-289.

Conclusions

Point 10: The conclusions should summarize the whole study and wrap up the story. The existing sections is just a final comment on the article.

Response 10: Thank you for your comment. We have now amended the conclusion in pages 15-16 lines 362-371, to read as:

In conclusion, technology apps are an effective media and are comparative with the traditional interventions for improving healthy food purchasing and nutrition-related outcomes among many diverse communities especially among younger adults and female users. The strengths of technology apps are attributed to its nature of low cost, wide reachability, and people have more time available to browse for online purchasing platforms especially during thisCOVID-19 pandemic. However, the need for continued engagement and certain complicated app features are the key factors that can limit its effectiveness. Therefore, apps development should be tailored with different degree of healthy food choices. There are definitely more room for development and testing, particularly in this ever vibrant app market, to establish its long-term efficacy.